# Upcycled high-strength aluminum alloys from scrap through solid-phase alloying

Tianhao Wang ●, Xiao Li ● ✉, Zehao Li ●, Tingkun Liu ●, Xiang Wang ●, Arun Devaraj ●, Cindy A. Powell ● & Jorge F. dos Santos ● ✉

Although recycling secondary aluminum can lead to energy consumption reduction compared to primary aluminum manufacturing, products produced by traditional melt-based recycling processes are inherently limited in terms of alloy composition and microstructure, and thus final properties. To overcome the constraints associated with melting, we have developed a solid-phase recycling and simultaneous alloying method. This innovative process enables the alloying of 6063 aluminum scrap with copper, zinc, and magnesium to form a nanocluster-strengthened high-performance aluminum alloy with a composition and properties akin to 7075 aluminum alloy. The unique nanostructure with a high density of Guinier-Preston zones and uniformly precipitated nanoscale $\eta'/Mg(CuZn)_2$ strengthening phases enhances both yield and ultimate tensile strength by >200%. By delivering high-performance products from scrap that are not just recycled but *upcycled*, this scalable manufacturing approach provides a model for metal reuse, with the option for on-demand upcycling of a variety of metallic materials from scrap sources.

Aluminum (Al) stands as the most extensively utilized non-ferrous metal globally[1,2], and the Al industry contributes 3% of all greenhouse gas emissions[3]. This is because the Hall–Héroult process used to produce primary Al from ore emits a significant amount of $CO_2$[4]. Using 100% scrap aluminum feedstocks could theoretically reduce energy and carbon emissions by up to 95% compared to primary aluminum production[5,6]. Nevertheless, this level of reduced environmental impact has not been achieved because scrap inevitably contains undesirable impurities, requiring the addition of primary Al in a process known as "purifying and diluting"[3,7–10]. Also, the traditional production of secondary Al through recycling involves a number of energy-intensive steps (Fig. 1a). The reliance on primary Al additions, combined with the requirement for high-temperature melting and other processing steps, results in a recycled product that still has relatively high energy and carbon intensity[3]. Friction extrusion, invented at The Welding Institute in the United Kingdom in 1993, is a solid-phase extrusion technique initially used to fabricate metal matrix composites[11]. In recent years, it has emerged as a promising method for recycling aluminum scraps[12,13], offering potential energy efficiency advantages over conventional processes due to its lower processing

temperatures and reduced number of steps. Friction extrusion offers the added advantage of rapidly breaking up heterogeneous feedstocks and then mixing and homogenizing them with the matrix material during processing, making it a scalable option for manufacturing composites with uniformly dispersed second phases[14–16].

In this work, we provide the report of friction extrusion as a viable solid-phase alloying method to directly upcycle Al scrap to high-performance Al wrought extrudate in a single step, with a much lower carbon footprint (Fig. 1b). For upcycling, 6063 Al scrap and alloying sources—copper (Cu) powder, zinc (Zn) powder, and ZK60 magnesium (Mg) ribbons—are first physically blended and then subjected to friction extrusion, removing the requirement for bulk melting to incorporate the alloying elements into the Al matrix. The thermomechanical nature of the friction extrusion process results in the production of a fine-grained aluminum microstructure with a high density of strengthening nanoclusters in the extruded product. Because this solid-phase process allows aluminum scrap to be upcycled into high-performance aluminum products without the need for either the addition of primary aluminum or melting, it has the potential to significantly reduce $CO_2$ emissions[17,18].

Pacific Northwest National Laboratory, Richland, WA, USA. ✉e-mail: xiao.li@pnnl.gov; jcpace911@gmail.com

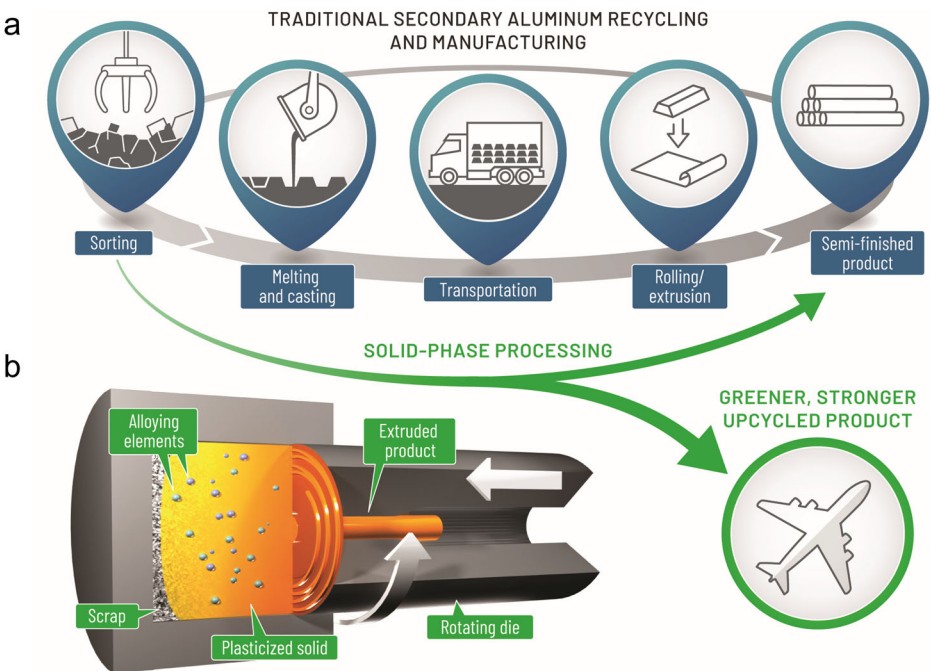

**Fig. 1 | Aluminum life cycle and the upcycle workflow. a** Traditional secondary Al recycling and manufacturing route: sorting, melting and casting, transportation, rolling/extrusion, and semi-finished products (rod, bar, and sheet). **b** Solid-phase processing directly recycles or upcycles scrap into extrudate via friction extrusion and reduces energy consumption by avoiding energy-intensive steps.

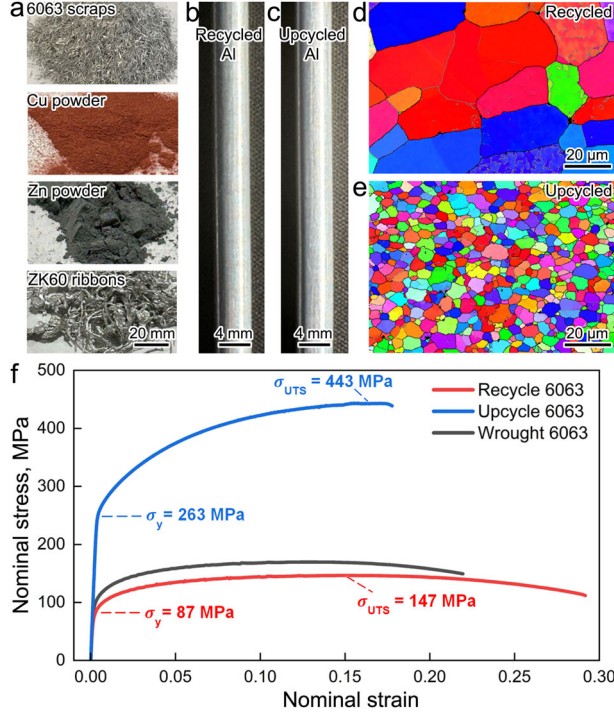

**Fig. 2 | Friction-extruded rods and their respective mechanical properties. a** Al scrap and alloying additions used in friction extrusion recycling and upcycling, from top to bottom: 6063 Al scrap, copper powder, zinc powder, and ZK60 magnesium alloy ribbons. **b** Recycled Al rod made only from cold-compacted 6063 Al scrap (the surface of the extruded rod was then manually polished). **c** Upcycled Al rod made from a mixture of 6063 Al scrap with alloying additions (the surface of the extruded rod was then manually polished). **d** EBSD-IPF showing the grain morphology of a recycled Al rod. **e** EBSD-IPF showing the grain morphology of the upcycled Al rod. **f** Stress-strain curves of friction-extruded samples: recycled Al rod and upcycled Al rod, compared with a commercial wrought 6063 Al alloy.

## Results

### Experimental results and mechanical properties

The precursor materials used in this work (Fig. 2a) include 6063 Al scrap, with alloying additions of Cu, Zn, and ZK60 Mg. The products of the upcycling process are benchmarked against a recycled 6063 alloy manufactured using the same friction extrusion approach. For the recycled alloy, cold-compacted 6063 scrap was friction extruded to produce a void-free, 5 mm diameter rod (Fig. 2b). For the upcycled product, the mixture of 6063 scrap and alloying additions were friction extruded under similar process conditions to produce a void-free rod (Fig. 2c) with a composition close to that of a standard 7075 Al alloy. The amount of each alloying addition in the mixture was calculated based on the chemical composition differences between 6063 scrap and the standard 7075 Al alloy, with details provided in the "Methods" section. It is important to note that in today's recycling industry, aluminum scrap is graded by its chemical composition defined by the Institute of Scrap Recycling Industries. This grading makes the current upcycling approach a feasible pathway to achieve the targeted compositions in the final upcycled alloys. The recycled rod has an average grain size of 43.1 μm, as shown in the electron backscatter diffraction-inverse pole figure (EBSD-IPF) map in Fig. 2d. The upcycled rod, on the other hand, shows pronounced grain refinement, with an average grain size of 7.7 μm (Fig. 2e). The evident grain refinement in the upcycled rod compared to the recycled rod is due to the reduced extrusion temperature in the upcycling process. However, the reason for the lower extrusion temperature in the upcycling process compared to the recycling process requires further investigation. A comparison of the tensile stress-strain curves for the recycled 6063 Al alloy, the upcycled 6063 Al alloy, and a conventional wrought 6063 Al alloy (Fig. 2f) highlights the pronounced increase in yield strength and ultimate tensile strength for the upcycled material. Specifically, upcycling results in a >200% increase in both the yield strength (from 87 MPa to 263 MPa) and the ultimate tensile strength (from 147 MPa to 443 MPa) when compared to the recycled alloy.

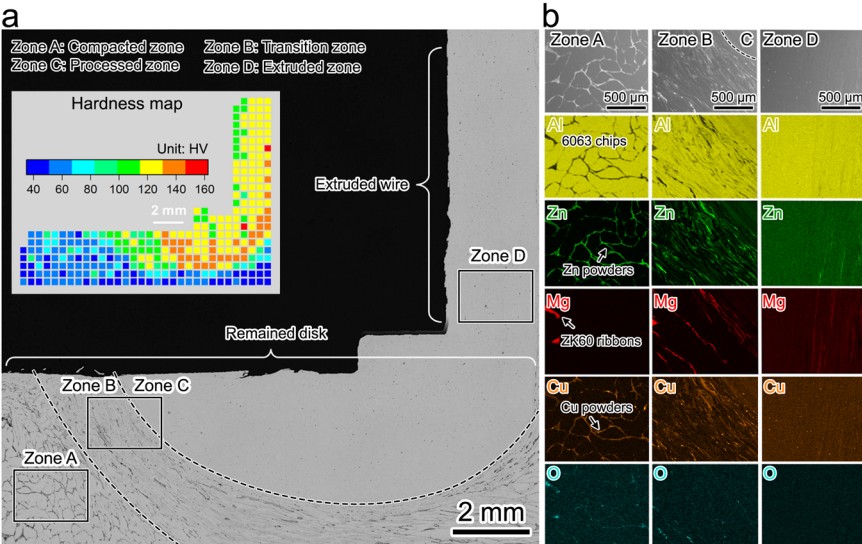

**Fig. 3 | Microstructural evolution during solid-phase alloying. a** Optical image displaying four distinct regions from bottom to top—Zone A: compacted scrap, powders, and ribbons. Zone B: deformed materials transition from compacted microstructure to processed microstructure. Zone C: mixed and alloyed microstructure. Zone D: extruded microstructure. Note the hardness map inset in (**a**), covering the span from Zone A to Zone D. **b** SEM and EDS mapping of Al, Zn, Mg, Cu, and O (from top to bottom) at Zone A, Zone B, and Zone D (from left to right).

## Microstructure evolution

Examining the microstructural evolution during the upcycling process provides insight into the origins of the differences in mechanical properties between recycled and upcycled materials. Optical microscopy of the longitudinal cross-section of the extruded rod and remaining disk (scrap/powders mixture) from the upcycling process (Fig. 3a) reveals four distinct processing zones—an unprocessed region consisting of the compacted scrap/alloying additions (Zone A), a transition region (Zone B), a processed region (Zone C), and the extruded rod (Zone D). Process temperatures increase exponentially from Zone A to Zone C, where the highest processing temperature is reached. The cooling process begins in Zone D. In the present study, no forced cooling was applied. Higher-magnification scanning electron microscopy (SEM) images, combined with energy dispersive spectroscopy (EDS) maps, from Zones A to D are also included in Fig. 3b. These images show that in Zone A, the 6063 Al scrap is surrounded by Zn powders, ZK60 Mg alloy ribbons, and Cu powders, which are tightly compacted. In Zone B, the alloying additions undergo severe shear deformation and are highly strained, elongated, and fractured. In Zone C, further refining promotes dispersion and diffusion of the alloying additions into the Al matrix before extrusion. In Zone D, the extruded product has similar microstructural features to those observed in Zone C. It is clear from these images that the additional alloying elements Zn, Mg, and Cu become progressively more homogeneously distributed in the microstructure as the material is processed from Zone A to B, to C, and finally to D. In other words, these alloying elements are refined and mixed into the Al matrix during the friction extrusion process, while the Al scrap was simultaneously consolidated into the extrudate[14,16]. These observations are supported by a hardness map (inset in Fig. 3a), which shows the variation in hardness that results from the above-described changes in microstructure. This type of microstructure evolution during friction extrusion has been reported in our previous work on the 7075 Al alloy system[19] and can be attributed to gradients in strain and temperature fields as the material is processed[20,21].

To confirm the homogeneity of the microstructure and chemistry of the upcycled and recycled rods, additional microscopy was conducted on both transverse and longitudinal cross-sections, with illustrative results provided in Fig. S1 (Supplementary Information Note 1). Hardness testing on the cross-sections of both recycled and upcycled rods reveals an average hardness of 43 HV for the recycled rod versus an average hardness of 116 HV for the upcycled rod (Fig. S2, Supplementary Information Section 2). To confirm that the upcycled product can meet performance requirements, a post-extrusion 7075-T6 heat treatment was performed on the as-upcycled sample. As shown in Fig. S3, the average hardness of the extruded rod increased from approximately 130 HV (Fig. S3a) to about 170 HV (Fig. S3b) after the T6 heat treatment, which is close to the standard hardness of 175 HV for 7075-T6.

## Phase identification and strengthening mechanism

Further microstructural characterization of the recycled and upcycled specimens pinpoints the reason for the substantial strengthening achieved as a result of the upcycling process. X-ray diffraction (XRD) (Fig. 4a) indicates that $Mg_2Si$ is the dominant precipitate phase in the recycled material. In the upcycled material, η'/$Mg(CuZn)_2$ is present in addition to $Mg_2Si$. High-angle annular dark-field scanning transmission electron microscopy (HAADF-STEM) reveals the presence of numerous Guinier–Preston (GP) zones along with several ordered precipitates within the Al matrix (Fig. 4b). A high-magnification bright field (BF) image, also shown in Fig. 4b, confirms that the structure of the ordered precipitates is consistent with that of the η' phase[22,23]. Interestingly, we also found evidence that the reaction occurs between pre-existing $Mg_2Si$ particles and dissolved alloying elements Cu and Zn. As shown in Fig. 4c, new phases (rich in Cu and Zn) form next to $Mg_2Si$ particles. High-resolution transmission electron microscopy reveals that the newly formed phases are amorphous Mg(ZnCu), with a crystalline η' phase/$Mg(CuZn)_2$ within the amorphous phase. More detailed information is provided in Fig. S4 (Supplementary Information Note 3). Three-dimensional atom maps of Al, Zn, Mg, and Cu distributions in the upcycled sample (Fig. 4d) reveal fluctuations in the Zn and Mg atom distributions within the Al matrix. The frequency distribution of the second nearest neighbor (2NN) distance (Fig. S5a) reveals the presence of clusters consisting of Zn, Mg, and Cu atoms. Figure 4e shows various crystallographic poles observed from the desorption map of the atom probe tomography (APT) dataset. The stacking of $(11\bar{1})$ atomic planes is clearly resolved in the Al map along the [111] crystallographic pole (Fig. 4f). The Mg, Zn, and Cu maps (Fig. 4g) show spherical (green arrow) and elongated (red arrow) precipitates on the

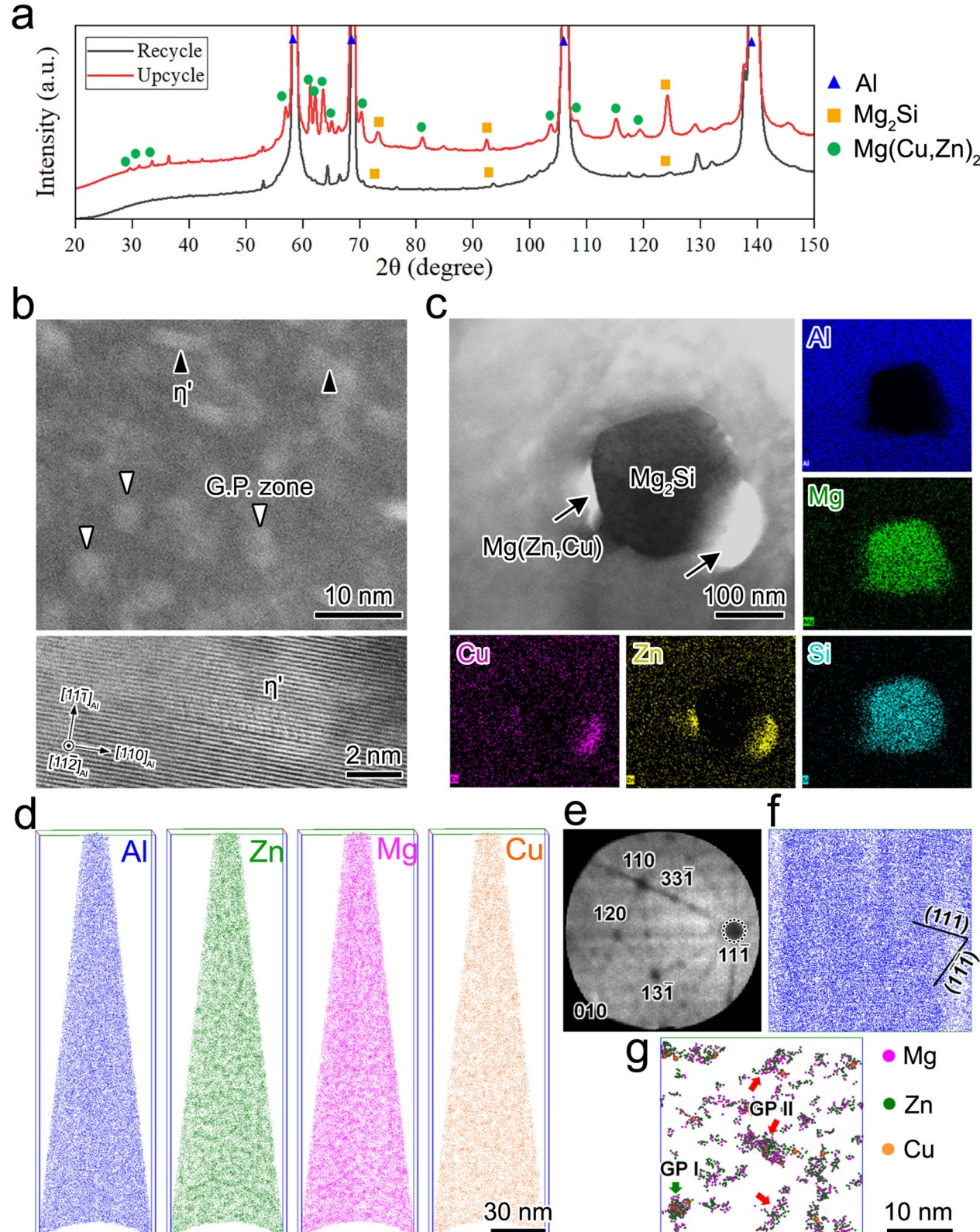

**Fig. 4 | Phase identification and microstructure of the recycled and upcycled Al via solid-phase processing. a** XRD analysis of the recycled and upcycled Al indicated that new Mg(CuZn)₂ phases were formed in the upcycling process. **b** HAADF and BF images of the upcycled specimen reveal the existence of the GP zone and η' phase nanoclusters. **c** Transmission electron microscopy and EDS analysis of pre-existed Mg₂Si particles reacting with newly dissolved Zn and Cu. They show potential reaction products next to pre-existing Mg₂Si particles. **d** 3D atom maps of Al, Zn, Mg, and Cu in the upcycled specimen. **e** The desorption map of the entire APT dataset with indexed crystallographic poles. **f, g** Sliced atom maps of Al, Zn, Mg, and Cu along the [11$\bar{1}$] pole of a selected volume with dimensions of $30 \times 30 \times 30\,\text{nm}^3$.

($11\bar{1}$) planes, which correspond to Guinier–Preston I (GPI) and GPII zones[24–26], respectively. The number density of the GP zones is calculated to be ~$1.6 \times 10^{24}$ m$^{-3}$ (Fig. S5b). These results provide strong evidence that solid-phase alloying is achieved during the upcycling process.

Typically, the development of GP zones within an Al matrix relies on two key factors: first, the matrix must be in a supersaturated solid solution state, usually achieved through a combination of solution heat treatment and rapid quenching. Second, there should be low-temperature artificial aging[27] or room temperature natural aging[28], during which GP zones nucleate and grow. However, subjecting the material to plastic deformation after attaining a supersaturated solid solution state can significantly expedite the aging process. This acceleration is attributed to dislocations generated during plastic deformation serving as nucleation sites for GP zone formation[29,30]. Recent research has demonstrated that continuous plastic deformation can facilitate the direct formation of GP zones without the need for further aging steps. For instance, the cyclic deformation of a supersaturated aluminum alloy at room temperature results in the dynamic precipitation of solute clusters (GP zones) by continuously introducing vacancies into the matrix[31]. Similar observations have been made in supersaturated magnesium alloys, where equal channel angular extrusion generates excess vacancies, leading to solute segregation and direct GP zone formation[32]. In the current study, the precursor materials are not in a supersaturated condition. Consequently, we hypothesize that two processes occur essentially simultaneously during the upcycling process: (1) dissolution of alloying additions into the matrix to create a supersaturated solid solution, and (2) formation of GP zones due to the continuous generation of vacancies through severe plastic deformation. It should be noted that applying the upcycling concept through the casting approach (also known as remelting) can introduce alloying elements into the Al matrix. However, multiple steps—including homogenization, quenching, extrusion, and post-extrusion heat treatment—are required after remelting to achieve the strengthening precipitates of $\eta'$/Mg(CuZn)$_2$. In contrast, these precipitates can be obtained in a single-step upcycling process in the current study.

The presence of oxygen, whether introduced as oxides in the feedstock or during the manufacturing process, is a significant concern and can pose major issues in the hot extrusion of aluminum chips[33]. However, in this study, oxides were barely present in the friction-extruded aluminum alloys. Microstructural analyses indicate that oxides introduced with the feedstock are broken apart, and oxygen is homogeneously distributed in the microstructure due to friction extrusion, reducing their potential to adversely affect product properties. Additionally, the relatively lower process temperatures and times used in friction extrusion decrease the degree of oxidation compared to hot extrusion. The EDS analyses provided in Fig. 3b show that before processing (Zone A), the surfaces of the feedstock particles are oxidized. As the alloying additions undergo shear deformation (Zone B), the oxides become strained, elongated, and fractured. In Zones C and D, oxygen-rich regions are no longer evident, indicating that the oxides have broken into smaller fragments and are distributed somewhat homogeneously throughout the microstructure. APT analysis of the upcycled sample shows uniform distributions of oxygen in the matrix with concentrations of less than 0.1 at.% (Fig. S6), indicating that there is a very low amount of uniformly distributed oxygen in the microstructure of an upcycled sample at the atomic scale. These results contrast with what occurs during conventional hot extrusion of aluminum chips, where the average oxygen concentration increases at each step of the process[34].

The disparity in strength between recycled and upcycled materials (Fig. 2f) arises from the observed microstructural variations that can activate various strengthening mechanisms, such as grain boundary strengthening, solid solution strengthening, dislocation

strengthening, and precipitate strengthening. Differences in grain size contribute to the strength increase, specifically with $\Delta\sigma_{gb}$ (grain boundary strengthening) values of 43.2 MPa for upcycled materials and 13.1 MPa for recycled materials (refer to Supplementary Information Section 4 for detailed calculations). Alloying content variations also play a significant role in the strength difference, leveraging both solid solution strengthening and precipitate strengthening mechanisms. In particular, $\Delta\sigma_{ss}$ (solid solution strengthening) values for upcycled and recycled materials are 46.8 MPa and 7.4 MPa, respectively (detailed calculations are included in Supplementary Information Section 5). The assessment of precipitate strengthening relies on the 3D atom probe analysis findings depicted in Fig. 4d. Notably, the presence of $\eta'$ precipitates is considered insignificant for strength calculations. However, the presence of GP zones has a significant effect. The 3D atom probe analysis provides details on the average size and volume fraction of GP zones in the upcycled specimen, with strength contributions ($\Delta\sigma_{Orowan}$) from GP zones amounting to 115.9 MPa. Note that the strength attributed to GP zones (coherent to the Al matrix) is anticipated to result from a dislocation shearing mechanism ($\Delta\sigma_{CS}$) rather than an Orowan dislocation bypassing mechanism ($\Delta\sigma_{Orowan}$). However, the calculation revealing $\Delta\sigma_{Orowan}$ to be smaller than $\Delta\sigma_{CS}$ implies that Orowan dislocation bypassing is the operative mechanism for GP zones in this case. A comprehensive description of the calculation is provided in Section 6 of the Supplementary Information. It is crucial to note that the recycled specimen lacks GP zones, making the determined $\Delta\sigma_{Orowan}$ from GP zones in the upcycled specimen an indicative measure for the precipitate strength contrast between the recycled and upcycled specimens. The variation in dislocation density does not significantly influence the strength difference, with ($\Delta\sigma_d$) (dislocation strengthening) values of 52.3 MPa for upcycled and 56.7 MPa for recycled specimens (refer to Supplementary Information Section 7 for detailed calculations). In total, theoretical calculations predict a yield strength improvement of 181 MPa in the upcycled aluminum, versus the recycled aluminum, in good agreement with the measured difference of 176 MPa (refer to Fig. 2f).

In summary, a high-performance Al alloy is obtained by upcycling 6063 scraps with copper, zinc, and magnesium in a single-step, solid-phase alloying process. The yield and ultimate tensile strength of the upcycled material is increased by >200% compared to the recycled material, due primarily to the formation of GP zones. These results demonstrate that low-strength and low-cost Al alloy scrap can be upcycled to high-strength and high-value Al alloy products by incorporating alloying elements via scalable solid-phase processing (friction extrusion), without the need to melt the precursor materials. Severe shear deformation imposed during friction extrusion refines the alloy additions and facilitates their uniform dispersion in the Al matrix, leading to the formation of fine GP zones and $\eta'$/Mg(CuZn)$_2$. Of larger significance, these results provide a demonstration of an approach to alloy design and manufacturing that creates value from waste, reduces the energy footprint and environmental impact of metals production, and offers a pathway to entirely new alloys and composites that cannot be produced by conventional melt-based processes.

## Methods

- *Base materials and tooling*

    6063 Al alloy scrap, Zn powders, Cu powders, and ZK60 Mg alloy ribbons were used as feedstock materials in this study. The chemical compositions of the 6063 Al alloy and ZK60 Mg alloy are provided in Table 1. A ZK60 Mg alloy was selected instead of pure Mg to avoid flammability issues. The 6063 chips and ZK60 ribbons have approximate lengths ranging from 1 to 5 mm and thicknesses from 0.1 to 1 mm, as shown in Fig. 2a. Both Cu and Zn powders are ~325 mesh, indicating a particle size of less than 44 μm. SEM images and EDS analysis of the Cu and Zn powders

are provided in Fig. S7 (Supplementary Information Section 8). The chemical composition of the target 7075 Al alloy is also listed in Table 1.

Friction extrusion tooling was made of H-13 tool steel. Spiral grooves were machined into the die face to facilitate material flow into the extrusion orifice, with a diameter of 5 mm. Die temperature was measured during processing using a type-K thermocouple (TC) spot-welded approximately 0.5 mm back from the die face.

**Table 1 | Nominal chemical composition of 6063 scrap, additional alloying powders and ribbons, mixed precursor, and standard 7075.[37]**

| Material | Chemical composition (wt. %) | | | | | | |
|---|---|---|---|---|---|---|---|
| | Al | Mg | Si | Zn | Cu | Fe | Zr |
| 6063 scraps | Bal. | 0.45 | 0.4 | 0.02 | 0.02 | 0.2 | - |
| Cu powders | - | - | - | - | 100 | - | - |
| Zn powders | - | - | - | 100 | - | - | - |
| ZK60 ribbons | - | Bal. | - | 5–5.5 | - | - | 0.57 |
| Mixed precursor | Bal. | 2.5 | 0.4 | 5.6 | 1.6 | 0.2 | 0.01 |
| Standard 7075 Al | Bal. | 2.1–2.5 | 0–0.5 | 5.6–6.1 | 1.2–1.6 | 0–0.5 | 0 |

- **Friction extrusion process**

  The manufacturing process for the recycled and upcycled rods is illustrated in Fig. 5a. First, precursor scrap and alloying elements were weighed out according to the chemical compositions shown in Table 1. Based on the 32 grams of 6063 scrap and the additives, the final weight percentages of Mg, Cu, and Zn in the upcycled material were approximately 2.5%, 1.6%, and 5.6%, respectively. Second, the precursor materials were loaded into a plastic bottle, and the bottle was sealed and placed on a roller mixer for 1 h. Third, the premixed precursor was loaded into a billet container and cold compacted with ~110 MPa using a hand press. Then, the liner with the pre-compacted precursor was loaded for friction extrusion into a ShAPE ™ machine manufactured by Bond Technologies. Critical friction extrusion parameters, such as starting and steady-state spindle speed and plunge speed, are listed in Table 2. Extrusion temperatures for both the recycling and upcycling processes are shown in Fig. 5b, c, respectively, along with the spindle speed and plunge speed. Additionally, based on the plunge depth and plunge speeds for both processes, it can be inferred that the friction extrusion time takes up to 5 min.

- **Sample preparation**

  Friction extrusion specimens were cut along the transverse cross-section and longitudinal cross-section using a diamond

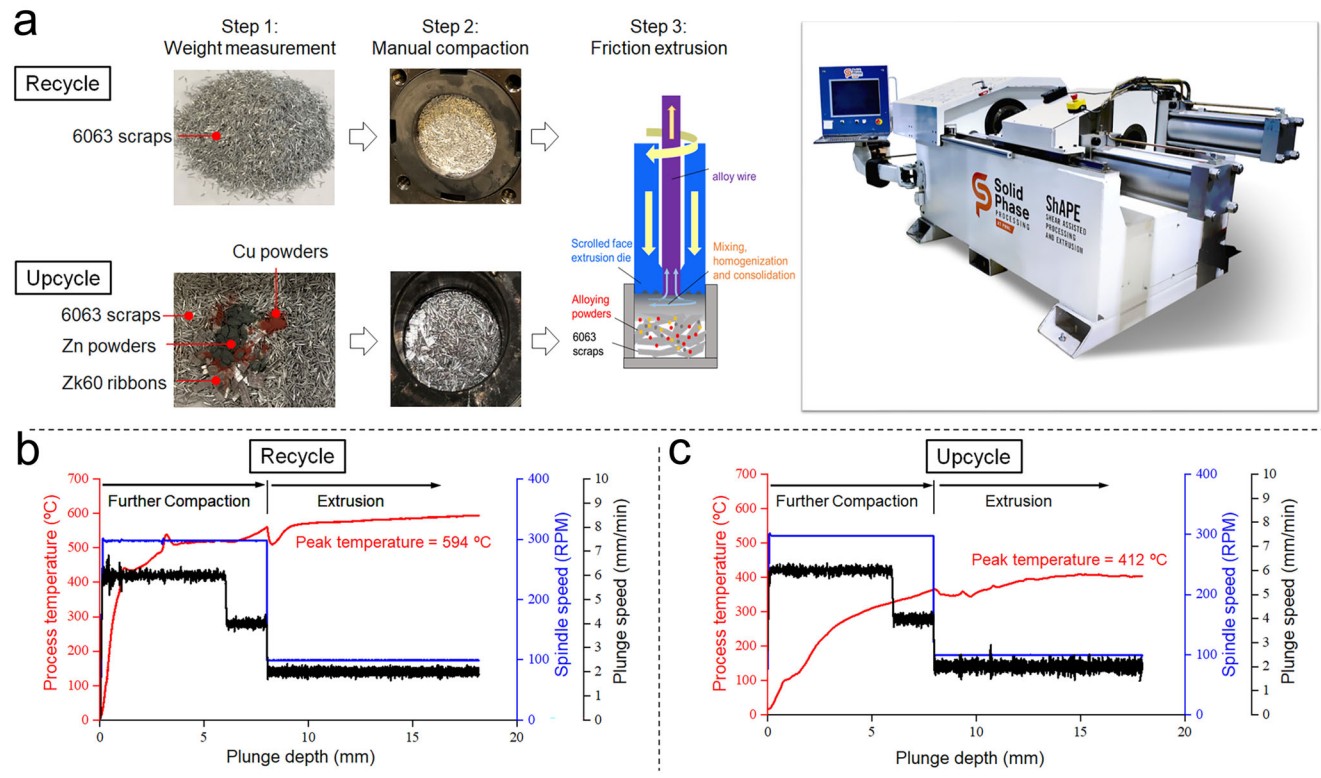

**Fig. 5 | Experiment procedures for both recycling and upcycling processes.** **a** The procedure steps include weighing chips and powders, manual compaction, and friction extrusion for both the recycling process (6063 scraps only) and the upcycling process (6063 scraps and alloying additives). The image shows the machine used for friction extrusion: ShAPE™. **b** The extrusion temperature, spindle speed, and plunge speed during recycling extrusion. **c** The extrusion temperature, spindle speed, and plunge speed during upcycle extrusion.

**Table 2 | Critical friction extrusion parameters for recycling and upcycling processes**

| Process | Starting material | Spindle speed (RPM) | Plunge speed (mm/min) |
|---|---|---|---|
| Recycle | 6063 Al alloy scrap | From 300 to 100 | From 6 to 4 and 2 |
| Upcycle | 6063 Al alloy scraps + Zn powders + Cu powders + ZK60 Mg alloy ribbons | From 300 to 100 | From 6 to 4 and 2 |

blade. Specimens for microstructural analysis were mounted in epoxy and polished to a final surface finish of 0.05 μm using colloidal silica.

Three mini-tensile specimens were machined from the 5 mm diameter extruded rods for both the recycled and upcycled materials. The round bar specimens had a 38.1 mm total length and 15.88 mm gauge length, following the ASTM-E8 standard.

A standard 7075-T6 post-extrusion heat treatment was applied to the upcycled sample, which included a solution heat treatment at 480 °C for 45 min, followed by water quenching, artificial aging at 120 °C for 24 h, and air cooling to room temperature.

- *Microstructural characterization*

  Optical microscopy was performed using an Olympus BX-51 fluorescence motorized microscope.

  SEM was performed using a Thermo Fisher Scientific Quanta 200 focused ion beam (FIB)-SEM outfitted with an Oxford Instruments X-ray EDS system for compositional analysis. Electron backscatter diffraction (EBSD) was done using a Thermo Fisher Scientific Apreo 2S SEM with an Oxford Instruments EBSD detector operating at 20 kV and a step size of 0.2 μm. The EBSD results were analyzed using ZtecCrystal EBSD processing software.

  Transmission electron microscopy (TEM) specimens were prepared by following the routine lift-out and thinning procedure of the FIB technique on a Thermo Fisher Scientific Quanta 200 FIB-SEM or a Thermo Fisher Scientific Helios 5 Hydra Dual Beam plasma focused ion beam milling SEM.

  TEM/STEM observations were performed on an FEI (now Thermo Fisher Scientific) Titan 80-300 Environmental Cs-corrected TEM equipped with an EDS system. HAADF and BF STEM images were captured on a JEOL GrandARM-300F operating at 300 kV with a convergence semi-angle of 29.7 mrad. The collection angles for HAADF-STEM were between 75 and 515 mrad.

  X-ray diffraction analysis was performed using a Rigaku D/Max Rapid II microdiffraction system. Diffraction data recorded on a two-dimensional image plate were integrated for diffraction angles between 20° and 150° using the manufacturer's software (2D Data Processing Software v.1.0, Rigaku, 2007).

  3D atom probe (3DAP) analysis was performed using a local electrode atom probe (CAMECA LEAP 6000 XR) in voltage pulse mode with a pulse fraction of 20% at a temperature of 40 K. Sharp needle-like specimens for the 3DAP analysis were prepared by the FIB lift-out and annular milling techniques using an FEI Helios 5 Hydra UX. The statistics and chemistry of precipitates in the selected volume without the Al poles were analyzed using a maximum separation algorithm in the AP Suite™ 6.3 software. The parameters of separation distances ($d_{max} = 0.58$), the minimum number of solutes ($N_{min} = 12$), envelop distances ($L = 0.58$), and erosion distances ($d_{erosion} = L$) were selected based on the nearest neighbor approach described elsewhere.[35]

- *Mechanical property characterization*

  Hardness values for the recycled and upcycled materials were measured on the cross-section using a Vickers microhardness tester at 200 g load, with a dwell time of 12 s.

  Tensile tests were carried out on an MTS mechanical testing machine using a constant rate of 0.08 mm/min along with digital image correlation analysis measuring deformation strain.

## Data availability

The authors declare that the data supporting the findings of this study are available within the paper and its supplementary information files. The source data underlying Figs. 2f, 3, 4, 5b, c, Supplementary Figs. 2, 3, 5a, b, 6b, c, 7a, b are provided as a Source data file. The Source data are also available at https://doi.org/10.6084/m9.figshare.27072622.v1[36].

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

## Acknowledgements

Pacific Northwest National Laboratory (PNNL) is a multiprogram national laboratory operated by Battelle for the DOE under Contract DEAC05-76RL01830. Current work was supported by the Laboratory Directed Research and Development program at PNNL as part of the Solid Phase Processing Science Initiative (SPPSI). The authors are grateful for the efforts of Anthony Guzman for preparing microstructural characterization specimens. This work was performed, in part, at the William R. Wiley Environmental Molecular Sciences Laboratory, a national scientific user facility sponsored by the U.S. Department of Energy, Biological and Environmental Research program and located at PNNL. This work was supported by the Laboratory Directed Research and Development program at PNNL as part of the SPPSI.

## Author contributions

J.F.S. and X.L. were the lead scientists of the study and proposed the core concept; X.L. and T.W. designed and fabricated the samples; T.W. and X.L. wrote the paper and addressed reviewers' comments; T.W. and X.L. analyzed experimental data and performed mechanical tests. Z.L. performed APT and related analysis; T.L. performed XRD, SEM, EBSD, and related analysis, X.W. performed TEM and related analysis. A.D. provided guidance for characterizations and writing. C.P. and J.F.S. funded and supervised the research. All authors contributed to the discussion.

## Competing interests

The authors declare no competing interests.
