## [Peer Review file · Nature Communications]

Upcycled High-Strength Aluminum Alloys from Scrap through Solid-Phase Alloying

Corresponding Author: Dr Xiao Li

Version 0:

Reviewer comments:

Reviewer #1

(Remarks to the Author)

In general this paper is very interesting and timely. Recycling by solid-state is a very interesting option that deserves strong research involvement. The paper deserves a consideration for publication in "Nature Communication" but should consider the following concerns:

1. The state of the art on friction extrusion is a bit weak. The reviewer is aware of the limitation in size of the journal but maybe a couple of sentences could be added to highlight the novelty compared to all that was done on this process.
2. Line 70 "standard 7075 composition": you should clarify how you can reach the composition of 7075 alloy. Do you need to perform a chemical analysis each time you have a pile of chips to compensate the composition and reach the one of 7075? The upscaling of this process is not clear in that sense.
3. Figure 2: it seems that the recycled Al had a different roughness (or surface finish) than the upcycled Al: can you please quantify this (if the picture impression is relevant) and provide some explanation of that (eventually in the supplementary as this might be considered as a second order effect).
4. Figure 2: provide a size distribution of the powder (maybe as supplementary).
5. Line 104: You say the process takes less than a minute but for what quantity of material? How does that compare to classical recycling in terms of efficiency?
6. An important aspect: I believe it would be fairer to compare the upcycled alloy not only to 6063 produced by friction surfacing but also to classical cast recycling with the same Mg, Cu and Mg additions (i.e. a "cast upcycling"). As the modification of the precipitation is an obvious consequence of the addition of these alloying elements. To confirm the interest of the friction extrusion we need to see if this process is really responsible for the unusual precipitation or if this could be reached by classical casting.
7. One important point: it looks like the authors did not perform any additional artificial aging after the extrusion process. Would that increase further the hardness like in common T4-> T6 transformation of the 7075 alloy?

Typo: Line 93: .. is reached. The cooling process begins in zone D.-> I assume this is what you meant?

Suggestion: line 232: ref to figure 2f

Reviewer #2

(Remarks to the Author)

The paper is very interesting and it has a good potential impact.

The papers are requested to extend the description of the state of the art.

Reviewer #3

(Remarks to the Author)

Dear authors,

First of all congratulation to the very interesting article.

Clarity and context:

I don't think that the overall language or gramma needs any improvement!

Nevertheless I do have some remarks:

Validity:

1. As aluminum scrap as well as ZK60 and Zn powder is used in the study I would assume a remarkable effect due to oxygen containing surface layers. As for instance stated in Solid state recycling of aluminum chips: Multi-technique characterization and analysis of oxidation, Materialia Volume 31, September 2023. Also an increased oxygen content could be a further reason for the strength increase! Could you comment on that, or on how you treated the scrap/powder to reduce/prevent the material from oxidation!

2. The "mixability" and "homogeneity" of the powder as well as scrap material is for sure an important factor for the homogeneity and properties of the extruded rod. Therefore, information regarding size distribution and shape/morphology is of high importance and should be stated in the publication.

3. The influence of the temperature on the extrusion process but of course also on the resulting microstructure evolution is of significant importance! As the use of thermocouples is mentioned in the publication, the results of these measurements as well as the temperature influence on the microstructure should be included.

4. In the Supplementary Material Table 1, the grain diameters for the upcycled and recycled material are mixed up!

Originality and Significance:

The publication provides an innovative approach for upcycling of aluminum alloy scrap with a reduced CO2 footprint!

Data & Methodology:

Beside of point 2 in the Validity chapter the methods are described in detail and are according scientific standards.

Conclusions:

The provided conclusion is solid and linked to the high resolution analytics.

Improvements:

If the suggestions made validity the article would be improved further.

References:

The references are appropriate even so there are some newer results from other groups available, which could be included.

Version 1:

Reviewer comments:

Reviewer #1

(Remarks to the Author)

Thank you for all these detailed answers. I am highly satisfied with the changes made to the manuscript.

Reviewer #3

(Remarks to the Author)

Dear authors, again, congratulations to the scientifically and technologically highly relevant paper. Due to the changes and replies I'd like to recommend the paper for publication!

made.

Dear Editor and Reviewers,

Thank you very much for your valuable questions and comments on our manuscript. Below, please find our responses. Replies are written in blue, and the revised sections in the manuscript and supplementary materials are highlighted.

REVIEWER COMMENTS

Reviewer #1 (Remarks to the Author):

In general this paper is very interesting and timely. Recycling by solid-state is a very interesting option that deserves strong research involvement. The paper deserves a consideration for publication in “Nature Communication” but should consider the following concerns:

Comment 1: The state of the art on friction extrusion is a bit weak. The reviewer is aware of the limitation in size of the journal but maybe a couple of sentences could be added to highlight the novelty compared to all that was done on this process.

Reply: Thanks for your suggestion. We included a brief overview of the history of friction extrusion and emphasized that this work marks the first-time use of friction extrusion for upcycling scrap through solid-phase alloying. The relevant content has been updated on Page 2 of the manuscript, as detailed below

- **Page 2:** Friction extrusion, invented at The Welding Institute in the United Kingdom in 1993, is a solid-phase extrusion technique initially used to fabricate metal matrix composites [11]. In recent years, it has emerged as a promising method for recycling aluminum scraps [12, 13], offering potential energy efficiency advantages over conventional processes due to its lower processing temperatures and reduced number of steps. Friction extrusion offers the added advantage of rapidly breaking up heterogeneous feedstocks and then mixing and homogenizing them with the matrix material during processing, making it a scalable option for manufacturing composites with uniformly dispersed second phases [14-16]. This letter provides the first report of friction extrusion as a viable solid-phase alloying method to directly upcycle Al scrap to high-performance Al wrought extrudate in a single step, with a much lower carbon footprint (**Fig. 1b**).

Comment 2: Line 70 “standard 7075 composition”: you should clarify how you can reach the composition of 7075 alloy. Do you need to perform a chemical analysis each time you have a pile of chips to compensate the composition and reach the one of 7075? The upscaling of this process is not clear in that sense.

Reply: Yes. Same as melting-based recycling, it is critical to ascertain the chemical composition of the precursor metal waste in order to achieve a desired composition by adjusting the alloying element to the appropriate content. Knowing the chemical composition of a batch of scrap is not difficult in today’s recycling industry, where scrap is classified into different grades by its chemical composition (impurity content) and source, then priced accordingly and traded as a commodity. The Institute of Scrap Recycling Industries (ISRI) has a standard specification (<https://www.isri.org/docs/default-source/scrap-metals-theft/suggested-recyclable-materials-with-photoscodes.pdf?sfvrsn=2>) to represent the quality and composition of the materials bought and sold in the industry. Aluminum extrusion scrap is one of the most abundant grades. For example, clean and unalloyed 6063 extrusion scrap containing 97% to 99% aluminum content is ISRI Coded as “TATA,” and aluminum 6061/7075 extrusion scrap is ISRI Coded as “TUTU.” Another common Al waste comes from the mixed post-consumer waste stream and is ISRI coded as “twich.” This waste consists of floated fragmentizer aluminum scrap derived from a wet or dry

media separation device, and the twitch aluminum scrap must be dry and not contain more than 1% maximum free zinc, 1% maximum free magnesium, and 1% maximum analytical iron. With knowledge of the chemical composition of the scrap, one can utilize friction extrusion to upcycle the scrap into a desired known alloy, or a customized new alloy, simply by adjusting the chemical composition. The final composition can be evaluated and validated using inductively coupled plasma mass spectrometry (ICP-MS). The relevant content has been updated on **pages 3 and 4** of the manuscript, as detailed below

- **Pages 3 and 4:** The amount of each alloying addition in the mixture was calculated based on the chemical composition differences between 6063 scrap and the standard 7075 Al alloy, with details provided in the Methods section. It is important to note that in today's recycling industry, aluminum scrap is graded by its chemical composition defined by the Institute of Scrap Recycling Industries. This grading makes the current upcycling approach a feasible pathway to achieve the targeted compositions in the final upcycled alloys.

Comment 3: Figure 2: it seems that the recycled Al had a different roughness (or surface finish) than the upcycled Al: can you please quantify this (if the picture impression is relevant) and provide some explanation of that (eventually in the supplementary as this might be considered as a second order effect).

Reply: Thanks for your comment. During friction extrusion, the roughness of the surface can be determined by (1) process parameters, (2) material composition, and (3) the wear extent of the tooling. In the current study, we have conducted recycle and upcycle experiments using the same process parameters. However, the difference in precursor materials between recycling and upcycling alters certain extrusion process variables, such as extrusion temperatures, as updated in **Figs. 5b and 5c**. The relevant content has been updated on **page 13** of the manuscript, as detailed below. To eliminate the influence of tooling wear, we repeated the recycling and upcycling experiments using brand-new tools. The extruded rods (recycled and upcycled rods) were then manually polished to remove the rough surface. As a result, the recycled and upcycled samples now exhibit a very similar surface finish, as updated in **Figs. 2b and 2c**. The relevant content has been updated on **page 5** of the manuscript, as detailed below.

Page 13: Extrusion temperatures for both the recycling and upcycling processes are shown in Figs. 5b and 5c, respectively, along with the spindle speed and plunge speed. Additionally, based on the plunge depth and plunge speeds for both processes, it can be inferred that the friction extrusion time takes up to 5 minutes.

Fig. 5: Experiment procedures for both recycling and upcycling process.

a. The procedure steps include weighing chips and powders, manual compaction, and friction extrusion. The image shows the machine used for friction extrusion: ShAPE™. **b.** The extrusion temperature, spindle speed, and plunge speed during recycle extrusion. **c.** The extrusion temperature, spindle speed, and plunge speed during upcycle extrusion.

Fig. 1: Friction-extruded rods and their respective mechanical properties.

a. Al scrap and alloying additions used in friction extrusion recycling and upcycling, from top to bottom: 6063 Al scrap, copper powder, zinc powder, and ZK60 magnesium alloy ribbons. **b.** Recycled Al rod made only from cold compacted 6063 Al scrap (the surface of the extruded rod was then manually polished). **c.** Upcycled Al rod made from a mixture of 6063 Al scrap with alloying additions (the surface of the extruded rod was then manually polished). **d.** EBSD-IPF showing the grain morphology of a recycled Al rod. **e.** EBSD-IPF showing the grain morphology of the upcycled Al rod. **f.** Stress-strain curves of friction-extruded samples: recycled Al rod and upcycled Al rod, compared with a commercial wrought 6063 Al alloy.

Comment 4: Figure 2: provide a size distribution of the powder (maybe as supplementary).

Reply: Thanks for your suggestion. In the current study, a total of four types of powder, chip, and ribbon are used. The 6063 chip and ZK60 ribbon have approximate lengths ranging from 1 to 5 mm and thicknesses from 0.1 to 1 mm, as updated in the Methods section (the relevant content has been updated on page 12 of the manuscript, as detailed below). Both Cu and Zn powders are ~325 mesh, with particle diameters of less than 44 μm . Additionally, the morphology and actual size can be seen in the SEM images in Fig. S7 (the relevant content has been updated in Section 8 of the supplementary materials, as detailed below).

- **Page 12:** The 6063 chips and ZK60 ribbons have approximate lengths ranging from 1 to 5 mm and thicknesses from 0.1 to 1 mm, as shown in Fig. 2a. Both Cu and Zn powders are ~325 mesh, indicating a particle size of less than 44 μm . SEM images and EDS analysis of the Cu and Zn powders are provided in Fig. S7 (Supplementary Material Section 8).
- **Supplementary page 7:** 8. Size and morphology of Cu and Zn powders
Both Cu and Zn powders are ~325 mesh, indicating particle diameters of less than 44 μm . The morphology and composition of the Cu and Zn powders are presented through SEM images and EDS analysis, as shown in Fig. S7.

Fig. S7. SEM and EDS analysis of a. Cu powder and b. Zn powder.

Comment 5: Line 104: You say the process takes less than a minute but for what quantity of material? How does that compare to classical recycling in terms of efficiency?

Reply: Firstly, the authors have removed the phrase "less than a minute" from the manuscript because it is not accurate. In the upcycling process described in this study, 32 grams of 6063 scraps and the other

additives were used. The overall friction extrusion process takes approximately 5 minutes. The relevant content has been updated on **page 6** and **page 13** of the manuscript, as detailed below.

- **Page 6:** In other words, these alloying elements are refined and mixed into the Al matrix during the friction extrusion process less than a minute, while the Al scrap was simultaneously consolidated into the extrudate [14, 16].
- **Page 13:** Based on the 32 grams of 6063 scrap and the additives, the final weight percentages of Mg, Cu, and Zn in the upcycled material were approximately 2.5%, 1.6%, and 5.6%, respectively.
- **Page 13:** Additionally, based on the plunge depth and plunge speeds for both processes, it can be inferred that the friction extrusion time takes up to 5 minutes.

Secondly, it is important to note that the primary purpose of this initial research is to demonstrate the feasibility of solid-phase alloying and upcycling using friction extrusion. Potentially, it reduces casting, homogenization, transportation (from casting plant to extrusion plant), and extrusion to a single-step recycling process. Therefore, significant efficiency improvement and energy saving could be achieved. However, we feel it's too early to use lab-scale machine data to estimate the efficiency, and we aren't prepared to compare this process with classical recycling until we complete techno-economic analysis. One reference that might help answer your question is the techno-economic analysis that was conducted for a similar solid-phase process that extrudes aluminum alloy solid billets into tubes (Techno-Economic Analysis for Shear Assisted Processing and Extrusion (ShAPE) of High-Strength Aluminum Alloys). The results of that analysis show that ShAPE can be 55% cheaper than conventional extrusion. Using unhomogenized billets and high-speed extrusion (195 mm/s) and removing the solution heat treatment stage increases savings for the ShAPE model 60% by cost and 85% by energy with respect to conventional extrusion.

Comment 6: An important aspect: I believe it would be fairer to compare the upcycled alloy not only to 6063 produced by friction extrusion but also to classical cast recycling with the same Mg, Cu and Mg additions (i.e. a “cast upcycling”). As the modification of the precipitation is an obvious consequence of the addition of these alloying elements. To confirm the interest of the friction extrusion we need to see if this process is really responsible for the unusual precipitation or if this could be reached by classical casting.

Reply: Thanks for the reviewer's suggestion. We completely agree that comparing the upcycled 6063 aluminum alloy using solid-phase alloying with the conventional remelting of 6063 scraps and additional alloying elements is more meaningful. However, the outcomes of the conventional remelting approach are already well understood. In detail, the precipitate phase can be formed by classical processes but the desired size and distribution in the wrought 7075 are not achievable by casting alone. In casting, the solute-segregated regions (Zn, Cu, Mg) are formed during solidification, and then the ingot must be homogenized by post-heat treatment to improve the uniformity of solute elements in the matrix. The as-cast and homogenized 7075 microstructure and EDS analysis are shown in Fig. 1 and Table. 1 of the paper: Park, S. Y., & Kim, W. J. (2016). Difference in the hot compressive behavior and processing maps between the as-cast and homogenized Al-Zn-Mg-Cu (7075) alloys. Journal of Materials Science & Technology, 32(7), 660-670. Even after homogenization, the sizes of the second-phase precipitates on the grain boundaries are much larger than the solid-phase alloying and upcycling samples of our study. To make wrought 7075 product, the homogenized cast ingot is then rolled or extruded, followed with heat treatments again to achieve the desired precipitation size for an optimized strength and ductility. The entire process has at least four thermal cycles: casting, homogenization, extrusion/rolling, and aging. With solid-phase alloying, precipitates of the same size and dispersion can be obtained within one thermal

cycle and at lower temperature, suggesting it requires much less energy input and emits less CO₂. The relevant content has been updated on **page 9** of the manuscript, as detailed below.

- **Page 9:** It should be noted that applying the upcycling concept through the casting approach (also known as remelting) can introduce alloying elements into the Al matrix. However, multiple steps—including homogenization, quenching, extrusion, and post-extrusion heat treatment—are required after remelting to achieve the strengthening precipitates of $\eta'/\text{Mg}(\text{CuZn})_2$. In contrast, these precipitates can be obtained in a single-step upcycling process in the current study.

Comment 7: One important point: it looks like the authors did not perform any additional artificial aging after the extrusion process. Would that increase further the hardness like in common T4-> T6 transformation of the 7075 alloy?

Reply: Correct. A post-extrusion heat treatment of the upcycled sample can further increase the mechanical properties. The additional experimental results regarding the post-extrusion heat treatment samples have been added to the paper. Specifically, a standard 7075-T6 heat treatment was conducted on the as-upcycled samples according to the following procedure: (1) solution heat treatment at 480 °C for 45 minutes, (2) water quenching, (3) artificial aging at 120 °C for 24 hours, and (4) air cooling to room temperature. It can be seen that the average hardness in the extruded rod was elevated from ~130 HV to ~170 HV after T6 heat treatment, which is very close to the hardness of 175 HV of standard 7075-T6. That said, we'd like to add that the optimized post-heat treatment procedure for a solid-phase alloyed material is yet to be determined, because our starting condition is neither cast nor rolled. (The relevant content has been updated on **pages 6 and 7** of the manuscript and in Fig. S3 of the **supplementary materials**, as detailed below).

- **Pages 6 and 7:** To confirm that the upcycled product can meet performance requirements, a post-extrusion T6 heat treatment was performed on the as-upcycled sample. As shown in **Fig. S3**, the average hardness of the extruded rod increased from approximately 130 HV (**Fig. S3a**) to about 170 HV (**Fig. S3b**) after the T6 heat treatment, which is close to the standard hardness of 175 HV for 7075-T6.
- **Supplementary page 2:** The hardness maps for extruded products in the as-upcycled condition and the upcycled+T6 condition are displayed in **Fig. S3a** and **S3b**, respectively.

Fig. S3. Optical image displaying the remained disk and extruded wire and hardness map for **a.** the as-upcycled condition and **b.** the upcycled + T6 condition.

Comment 8: Typo: Line 93: . is reached. The cooling process begins in zone D.-> I assume this is what you meant? Suggestion: line 232: ref to figure 2f.

Reply: Thank you very much for the corrections. (The relevant content has been updated on **page 6** and **page 12** of the manuscript, as detailed below.)

- **Page 6:** The cooling process begins in Zone D.
- **Page 12:** ... (refer to **Page 5: f**)

Reviewer #2 (Remarks to the Author):

Comment 1: The paper is very interesting, and it has a good potential impact. The papers are requested to extend the description of the state of the art.

Reply: Thank you for your suggestion. We added a short description of friction extrusion and highlighted the fact that this work is the first-time use of friction extrusion to upcycle scrap via solid-phase alloying. The relevant content has been updated on **page 2** of the manuscript, as detailed below.

- **Page 2:** Friction extrusion, invented at The Welding Institute in the United Kingdom in 1993, is a solid-phase extrusion technique initially used to fabricate metal matrix composites [11]. In recent years, it has emerged as a promising method for recycling aluminum scraps [12, 13], offering potential energy efficiency advantages over conventional processes due to its lower processing temperatures and reduced number of steps. Friction extrusion offers the added advantage of rapidly breaking up heterogeneous feedstocks and then mixing and homogenizing them with the matrix material during processing, making it a scalable option for manufacturing composites with uniformly dispersed second phases [14-16]. This letter provides the first report of friction extrusion as a viable solid-phase alloying method to directly upcycle Al scrap to high-performance Al wrought extrudate in a single step, with a much lower carbon footprint (**Fig. 1b**).

Reviewer #3 (Remarks to the Author):

Dear authors, First of all congratulation to the very interesting article. Clairity and context: I don't think that the overall language or gramma needs any improvement! Nevertheless, I do have some remarks:
Validity:

Comment 1: As aluminum scrap as well as ZK60 and Zn powder is used in the study I would assume a remarkable effect due to oxygen containing surface layers. As for instance stated in Solid state recycling of aluminum chips: Multi-technique characterization and analysis of oxidation, Materialia Volume 31, September 2023. Also an increased oxygen content could be a further reason for the strength increase! Could you comment on that, or on how you treated the scrap/powder to reduce/prevent the material from oxidation!

Reply: Thanks for your comments. We have added the critical and relevant reference concerning oxidation during recycling to the manuscript. The distribution of oxygen (O) has been included in **Fig. 3** on **page 7** of the manuscript, as described below. Additionally, we have updated the description of oxidation during friction extrusion in the current study. Specifically, based on the EDS analysis shown in **Fig. 3b**, it is evident that oxygen initially surrounds the 6063 Al scrap in Zone A due to the presence of Zn and Cu powders covered by oxides. In Zone B, as the alloying additions undergo shear deformation, the oxides become strained, elongated, and fractured. In Zones C and D, there are no evident oxygen-rich regions, indicating that the oxides have fragmented into smaller pieces and distributed homogeneously throughout the microstructure (the relevant content has been updated on **pages 9 and 10** of the manuscript, as detailed below). Furthermore, APT analysis of the upcycled sample reveals uniform distribution of oxygen within the matrix at concentrations of less than 0.1 at.% (**Fig. S6**), indicating that there is only very low uniformly distributed oxygen in the microstructure of upcycled samples at the atomic scale (the relevant content has been updated in the **supplementary materials**, as detailed below).

Fig. 3: Microstructural evolution during solid-phase alloying.

a. Optical image displaying four distinct regions from bottom to top—Zone A: compacted scrap, powders, and ribbons. Zone B: deformed materials transition from compacted microstructure to processed microstructure. Zone C: mixed and alloyed microstructure. Zone D: extruded microstructure. Note the hardness map inset in **Fig. 3a**, covering the span from Zone A to Zone D. **b.** SEM and EDS mapping of Al, Zn, Mg, Cu, and O (from top to bottom) at Zone A, Zone B, and Zone D (from left to right).

- **Pages 9 and 10:** The presence of oxygen, whether introduced as oxides in the feedstock or during the manufacturing process, is a significant concern and can pose major issues in the hot extrusion of aluminum chips [33]. However, in this study, oxides were barely present in the friction-extruded aluminum alloys. Microstructural analyses indicate that oxides introduced with the feedstock are broken apart, and oxygen is homogeneously distributed in the microstructure due to friction extrusion, reducing their potential to adversely affect product properties. Additionally, the relatively lower process temperatures and times used in friction extrusion decrease the degree of oxidation compared to hot extrusion. The EDS analyses provided in **Fig. 3b** show that before processing (Zone A), the surfaces of the feedstock particles are oxidized. As the alloying additions undergo shear deformation (Zone B), the oxides become strained, elongated, and fractured. In Zones C and D, oxygen-rich regions are no longer evident, indicating that the oxides have broken into smaller fragments and are distributed somewhat homogeneously throughout the microstructure. APT analysis of the upcycled sample shows uniform distributions of oxygen in the matrix with concentrations of less than 0.1 at.% (**Fig. S6**), indicating that there is a very low amount of uniformly distributed oxygen in the microstructure of an upcycled sample at the atomic scale. These results contrast with what occurs

during conventional hot extrusion of aluminum chips, where the average oxygen concentration increases at each step of the process [34].

- **Supplementary page 4:** Fig. S6a shows the 3D atom maps of O, AlO, and Al₂O ions from the atomic probe tomography (APT) mass-to-charge spectra of the upcycled sample. The O⁺, AlO²⁺, AlO⁺, and Al₂O⁺ peaks are clearly detected in the corresponding mass-to-charge spectrum (Fig. S6b). Fig. S6c shows the 1D compositional profiles along the arrow direction of a selected volume of 25 × 25 × 100 nm³ in the inset. All oxide species detected are uniformly distributed in the matrix with concentrations less than 0.1 at.%.

Fig. S6. APT results from the upcycled sample. **a.** 3D atom maps of O, AlO, and Al₂O, **b.** mass-to-charge spectrum, and **c.** 1-D compositional profiles along the arrow direction of a selected volume of 25 × 25 × 100 nm³ in the inset.

Comment 2: The “mixability” and “homogeneity” of the powder as well as scrap material is for sure an important factor for the homogeneity and properties of the extruded rod. Therefore, information regarding size distribution and shape/morphology is of high importance and should be stated in the publication.

Reply: Thanks for your comments. We agree that the thorough mixing of scrap and powder before friction extrusion is crucial. In this study, we used a roller mixer to blend the precursors before upcycling friction extrusion. While we cannot precisely quantify the mixing quality of post-roller mixing, experimental observations after friction extrusion indicate that the process is not overly sensitive to the size or morphology of the feedstock materials. This is evident from the successful alloying of both large Mg alloy ribbons and small Cu and Zn powders into the aluminum matrix, as shown in Fig. 3. The relevant content has been updated on page 12 of the manuscript and in the supplementary materials, as detailed below.

- **Page 12:** The 6063 chips and ZK60 ribbons have approximate lengths ranging from 1 to 5 mm and thicknesses from 0.1 to 1 mm, as shown in Fig. 2a. Both Cu and Zn powders are ~325 mesh,

indicating a particle size of less than 44 μm . SEM images and EDS analysis of the Cu and Zn powders are provided in Fig. S7 (Supplementary Material Section 8).

- **Supplementary page 7:** *8. Size and Morphology of Cu and Zn powders*
Both Cu and Zn powders are ~ 325 mesh, indicating particle diameters of less than 44 μm . The morphology and composition of the Cu and Zn powders are presented through SEM images and EDS analysis, as shown in Fig. S7.

Fig. S7. SEM and EDS analysis of **a.** Cu powder and **b.** Zn powder.

Comment 3: The influence of the temperature on the extrusion process but of course also on the resulting microstructure evolution is of significant importance! As the use of thermocouples is mentioned in the publication, the results of these measurements as well as the temperature influence on the microstructure should be included.

Reply: We agree with the reviewer. To accommodate this suggestion, the relevant content has been updated in the manuscript. Specifically, the updates can be found on page 4, as detailed below. Additionally, the temperature history for producing both recycled and upcycled samples has been added in Fig. 5, with the corresponding content updated on page 13, as detailed below.

- **Page 4:** The evident grain refinement in the upcycled rod compared to the recycled rod is due to the reduced extrusion temperature in the upcycling process, as shown in Figs. 5b and 5c. However, the reason for the lower extrusion temperature in the upcycling process compared to the recycling process requires further investigation.
- **Page 13:** Extrusion temperatures for both the recycling and upcycling processes are shown in Figs. 5b and 5c, respectively, along with the spindle speed and plunge speed.

Fig. 5: . Experiment procedures for both recycling and upcycling process

a. The procedure steps include weighing chips and powders, manual compaction, and friction extrusion. The image shows the machine used for friction extrusion: ShAPE™. **b.** The extrusion temperature, spindle speed, and plunge speed during recycle extrusion. **c.** The extrusion temperature, spindle speed, and plunge speed during upcycle extrusion.

Comment 4: In the Supplementary Material Table 1, the grain diameters for the upcycled and recycled material are mixed up!

Reply: Thank you for pointing out the error. The content in supplementary S-Table 1 has been updated.

• **Supplementary page 5:**

S-Table 1. Grain boundary strengthening contributions based on grain size.

Condition	Grain diameter, average (um)	Hall-Petch coefficient (MPa·m ^{1/2})	$\Delta\sigma_{gb}$ (MPa)
Upcycled	7.7	0.12	43.2
Recycled	43.1	0.086	13.1

Comment 5: References: the references are appropriate even so there are some newer results from other groups available, which could be included.

Reply: Thank you for the suggestion. New results on solid-state recycling have been updated on pages 9 and 10 of the manuscript, as detailed below.

- **Pages 9 and 10:** The presence of oxygen, whether introduced as oxides in the feedstock or during the manufacturing process, is a significant concern and can pose major issues in the hot extrusion of aluminum chips [33]. However, in this study, oxides were barely present in the friction-extruded aluminum alloys. Microstructural analyses indicate that oxides introduced with the feedstock are

broken apart, and oxygen is homogeneously distributed in the microstructure due to friction extrusion, reducing their potential to adversely affect product properties. Additionally, the relatively lower process temperatures and times used in friction extrusion decrease the degree of oxidation compared to hot extrusion. The EDS analyses provided in **Fig. 3b** show that before processing (Zone A), the surfaces of the feedstock particles are oxidized. As the alloying additions undergo shear deformation (Zone B), the oxides become strained, elongated, and fractured. In Zones C and D, oxygen-rich regions are no longer evident, indicating that the oxides have broken into smaller fragments and are distributed somewhat homogeneously throughout the microstructure. APT analysis of the upcycled sample shows uniform distributions of oxygen in the matrix with concentrations of less than 0.1 at.% (**Fig. S6**), indicating that there is a very low amount of uniformly distributed oxygen in the microstructure of an upcycled sample at the atomic scale. These results contrast with what occurs during conventional hot extrusion of aluminum chips, where the average oxygen concentration increases at each step of the process [34].